# Autoimmune Glial Fibrillary Acidic Protein (Gfap) Astrocytopa-Thy Accompanied with Reversible Splenial Lesion Syndrome (RESLES): A Case Report and Literature Review

**DOI:** 10.3390/brainsci13040659

**Published:** 2023-04-14

**Authors:** Siting Wang, Jianlan Yuan, Jingli Liu

**Affiliations:** Department of Neurology, The First Affiliated Hospital of Guangxi Medical University, Nanning 530021, China

**Keywords:** GFAP, astrocytopathy, reversible splenial lesion syndrome, autoantibody

## Abstract

Background: Autoimmune glial fibrillary acidic protein (GFAP) astrocytopathy usually presents as meningoencephalomyelitis. Many patients developed flu-like symptoms preceding the neurologic symptoms. Reversible lesion in the splenium of the corpus callosum (SCC) is a clinical and radiological syndrome secondary to many kinds of etiologies, including infections, which is termed RESLES. Case presentation: we reported a case developing irregularly high fever, both temporal pain, low limbs fatigue with frequent urination admitted to our neurology department. CSF test showed GFAP-IgG positive, elevated WBC counts and protein, with low glucose and chlorine, while MRI showed a reversible lesion on SCC, leading us to diagnose autoimmune GFAP autocytopathy accompanied with RESLES. The boy had significantly improved after anti-virus and steroids therapy. Discussion: Autoimmune GFAP autocytopathy accompanied with RESLES is rarely seen, and pathogenesis for the co-existence has not been clarified. Autoimmune GFAP autocytopathy and RESLES are both related to viral infection. Our case covered infectious symptoms and improved after antiviral treatment, suggesting virus infection may perform a key role in pathogenesis.

## 1. Introduction

Autoimmune glial fibrillary acidic protein (GFAP) astrocytopathy is a spectrum of autoimmune inflammatory central nervous system (CNS) disorders that were first defined by Fang et al. (Mayo Clinic) in 2016 [1]. GFAP is a specific intermediate fibrin of astrocytes. The disruption of astrocytes causing by GFAP-specific CDT cells mediated cellular immunity leads to inflammatory cells accumulate, and allow autoimmune attacks spreading to healthy nerve tissues, manifesting as encephalitis and meningitis, followed by myelitis and optic neuritis. CSF usually shows a specific GFAP antibody, and MRI reveals abnormal hyperintensity on T2-weighted images and gadolinium enhancement on T1. Previous studies reveal 30–40% of patients developed flu-like symptoms before neurologic symptoms [2].

Reversible splenial lesion syndrome (RESLES) is a clinical and radiological syndrome, patients usually develop headache, altered consciousness and seizure, followed by mental abnormality, focal neurological deficit, personality change, and visual disturbance [3]. Current studies indicate infections, epileptic/antiepileptic factors, or metabolic diseases cause SCC myelin sheath edema, leading to abnormal signal on MRI, and lesions can totally disappeared or significantly shrink during follow-up.

Autoimmune GFAP autocytopathy accompanied with RESLES is rarely seen, only three cases have reported the same situation, but they are different in etiology, clinical, and MRI features [4,5,6]. The relationship between autoimmune GFAP astrocytopathy and RESLES remains unclear. Here, we report a case of autoimmune GFAP astrocytopathy coexist with RESLES, and review the previous literature, to probe into the possible link between autoimmune GAFP autocytopathy and RESLES.

## 2. Case Report

A 16-year-old boy became irregularly fever, both temporal pain, low limbs fatigue with frequent urination since 28 October 2021. The maximum body temperature reached 40.5 °C. Urinary output was normal. The patient’s first visit to the local hospital, initial blood testing revealed hypokalemia, hyponatremia, and hypochloremia (K^+^ 3.19 mmol/L, Na^+^ 132.9 mmol/L, Cl^−^ 94.7 mmol/L). A chest computed tomography (CT) scan was performed but showed no significant findings. The local hospital gave the treatment of Amoxicillin and antipyretic for few days, but he still felt ill.

A total of 6 days after onset, on 3 November 2021, he turned to a higher-level hospital for further evaluation and treatment. Lumbar puncture showed CSF ICP 350 mmH_2_O, glucose 2.25 mmol/L, chlorine 118.2 mmol/L, and protein 1.27 g/L. WBC counts were normal. Cerebral MRI demonstrated an abnormal signal on SCC (Figure 1). Based on these results, he was considered as meningoencephalomyelitis, and was given ceftriaxone to anti-infective. However, there was no improvement in the condition.

A total of 9 days after onset, on 6 November 2021, he was admitted to the infectious department of our hospital. Physical examination at hospital admission revealed fever, headache, mild weakness of muscle force (MRC grade 5), mild neck rigidity, mild misnicturition, and frequent urination, with weakened low limbs deep reflect and suspicious positive of Babinski’s sign without other meningeal syndrome and cranial nerves dysfunction.

Laboratory examinations indicated normal serum blood cell counts and the inflammatory work-up (CRP, ESR). No infectious agents were found including hepatitis virus, HIV virus, EB virus, cytomegalovirus, plasmodium, bacterial, and fungus by routine serum microbiological studies. Antinuclear antibody and rheumatoid factor were both negative. CSF analysis clued an increase in WBC counts (20 × 10^6^/L), high intracranial pressure, hypoglycorrhachia, and hyperproteinorachia (ICP 80 drops per minutes, Glu 1.93 mmol/L, Pro 1.27 g/L), CSF adenosine deaminase (ADA) was 8.50 U/L. Infection agents in CSF were not found by general smear and culture. No remarkable abnormality among chest CT and abdominal ultrasonography.

According to the results described above, he was primarily diagnosed with intracranial infections, and was given treatment by ceftriaxone and trimethoprim-sulfamethoxazole to anti-bacterial infection lasted for 5 days, but he was still symptomatic. For further diagnosis and treatment, he then transferred to our neurology department on 10 November 2021. Thinking there is no sufficient evidence of bacterial infection, we stopped anti-bacterial infection and starting with acyclovir to antivirus on 11 November 2021.

A repeated spinal tap was performed on 11 November 2021, which showed increased intracranial pressure, CSF glucose was still below normal contents, while CSF protein and chloride was lower than before and still abnormal (ICP 330 mmH2O, Glu 2.14 mmol/L, Cl^−^ 119.50 mmol/L, Pro 1.05 g/L). WBC counts increased to 500 × 10^6^/L. CSF screen using next-generation sequencing (NGS) indicated no infection. The CSF test of neural auto-antibodies revealed positive anti-GFAP antibodies (antibody titer 1:32), while the serum test was negative. CSF Ig electrophoresis revealed positive IgG oligoclonal bands (≥2 bands). A second cerebral MRI was performed on 13 November 2021 showing totally disappeared on SCC lesion (Figure 2).

Based on the combination of clinical features, CSF test, and cerebral MRI changes, we finally diagnosed the patient with autoimmune GFAP astrocytopathy accompanied with RESLES. On 12 November 2021, the second day of antiviral treatment, his body temperature returned to normal without headache. However, difficulty and frequency of urination still persisted. We then gave a high-dose intravenous steroid starting on 15 November 2021 at 1 g/day for 4 days followed by 0.5 g/day for 1 day and 0.25 g/day for 3 days. Rechecked spinal tap on 17 November 2021 revealed CSF WBC counts 240 × 10^6^/L, normal CSF protein, glucose, and chloride (Pro 0.45 g/L, Glu 3.61 mmol/L, Cl^−^ 128.00 mmol/L). On 22 November 2021, the patient was discharged from hospital with oral corticosteroids and go home to outpatient follow-up. During follow-up, his neurological symptoms were completely recovered within 3 months. Steroid therapy was progressively tapered over 6 months until complete withdrawal. A lumbar puncture was performed on 11 May 2022 before steroid withdrawal showed CSF WBC counts for 16 × 10^6^/L, normal CSF protein, glucose, and chloride levels. All the CSF results during hospitalization and outpatient visits are listed in Table 1.

## 3. Discussion

Autoimmune GFAP astrocytopathy is a spectrum of immunotherapy-responsive autoimmune inflammatory CNS disorders, with GFAP-IgG detected in CSF. RESLES is a class of diseases caused by infectious or non-infectious factors presenting as mild encephalitis/encephalopathy with a reversible splenial lesion on brain MRI. Autoimmune GFAP astrocytopathy accompanied with RESLES is rare and only three cases have been reported so far. To our knowledge, our case is the first reported in China.

Generally, typical features of autoimmune GFAP astrocytopathy present as a linear, radial perivascular enhancement pattern on brain MRI, and central longitudinally extensive enhancement pattern on spinal cord MRI [7]. RESLES is typically classified into two patterns on MRI: type I is an isolated lesion on SCC, type II is a lesion in SCC expending to callosal fibers, cerebral white matters or anterior portion of corpus callosum. No matter type I or type II, lesions can be significantly shrunk or totally disappeared within a month, accompanied with the relief of symptoms [8]. In the MRI of our case, the lesion was located in the SCC, manifested a hypointense signal on T1WI and hyperintense on T2WI and DWI, reversibly, which is similar to type I RESLES, but unlike the typical autoimmune GFAP astrocytopathy.

We made a brief review of the three GFAP-RESLES cases (Table 2). One is a 10-year-old boy, with a comparatively similar MRI characteristic as our case, an isolated and reversible lesion on SCC. The boy had fever, heachache and vomiting for two days before admission, then the neurological symptoms. No possible pathogens were found in his CSF. The boy did not receive antiviral therapy but improved significantly after Ig IV infusion [5]. Another case is an adult patient whose abnormal signals on MRI spread to white matter and meninges, not only located in SCC. The patient also suffered transient fever before neurological symptoms onset. No possible pathogens were found in CSF either. In hospital, she was first treated with acyclovir, amoxicillin, and cephalosporin, but became worse, which is different from our case. Then, they give the treatment of high-dose corticosteroids and her condition improved [4]. The one left of the cases was confirmed to be coexisted with primary central nervous system lymphoma, MRI demonstrated multiple abnormal enhancement lesions in bilateral basal ganglia and around the third ventricle, and reversible lesion on SCC, also different with our case. He never had any flu-like symptoms such as fever [6].

The previous 3 cases did not delve into the mechanisms of the two diseases co-occurring, so the underlying mechanism of autoimmune GFAP austrocytopathy with RESLES is still unclear. Based on previous studies, we speculate there may be two possible explanations for the disease occurring. One explanation is that virus infection may perform a key role in pathogenesis, since autoimmune GFAP astrocytopathy and RESLES can both be secondary to virus infection. When pathogens invade, the body’s inflammatory response to pathogens may activate cellular immunity mediated by GFAP-specific CDT cells [9,10,11,12,13,14]. CDT cells attack astrocytes then release chemokine which leads to accumulation of inflammatory cells, allowing autoimmune attacks to spread to healthy nerve tissues and cause cellular edema [15,16,17]. Due to the high water content and insufficient self-protection, cellular edema is more likely to occur on SCC. Distinctively, cellular edema of SCC belongs to intra-myelin sheath edema, which does not damage fibers on SCC, so the lesions are reversible [18]. Another possible explanation is that the reversible lesion on SCC may be secondary to autoimmune GFAP astrocytopathy, since several articles have reported cases of RESLES with some auto-immune diseases [19,20]. However, regretfully, no research proves that GFAP antibodies can result in reversible SCC lesion so far. Since we have not further studied the internal relationship between these two in pathology, we can only explain the relationship clinically; thus, in-depth research and more cases are essential to correctly illustrate the inner relations. The case we reported, the boy covered flu-like symptom with abnormal high fever before neurological symptoms onset, his body temperature dropped to normal after antiviral therapy for two days, with lesion in SCC disappeared, demonstrate viral infection is the common pathogenesis basis of the two disease. Of course, viral infection is the basic, but not the only pathogenic mechanism, and there should be secondary autoimmune response to virus infection.

Treatments of autoimmune GFAP autocytopathy in acute stage mainly depends on high-dose corticosteroids, other treatments, including intravenous immunoglobulin (Ig IV) and plasma exchange. Treatment in remission is the maintenance of oral steroids and immunosuppressant. In both our case and the previous cases, patients had significant improvement after given high-dose corticosteroids or Ig IV infusion, confirmed that hormone therapy and Ig IV infusion are still effective way for autoimmune GFAP astrocytopathy accompanied with RESLES. In addition, since viral infection is the common basis of both autoimmune GFAP astrocytopathy and RESLES, antiviral therapy may be helpful for patients with flu-like symptoms, but further studies are still needed to verify.

## 4. Conclusions

Autoimmune GFAP astrocytopathy accompanied with RESLES is rare, its clinical features, and CSF and MRI characteristics are consistent with autoimmune GFAP astrocytopathy and RESLES. The pathogenesis for the co-existence has not been clarified. Our case covered infectious symptoms and improved after antiviral treatment, suggesting virus infection may perform a key role in pathogenesis.

## Figures and Tables

**Figure 1 brainsci-13-00659-f001:**
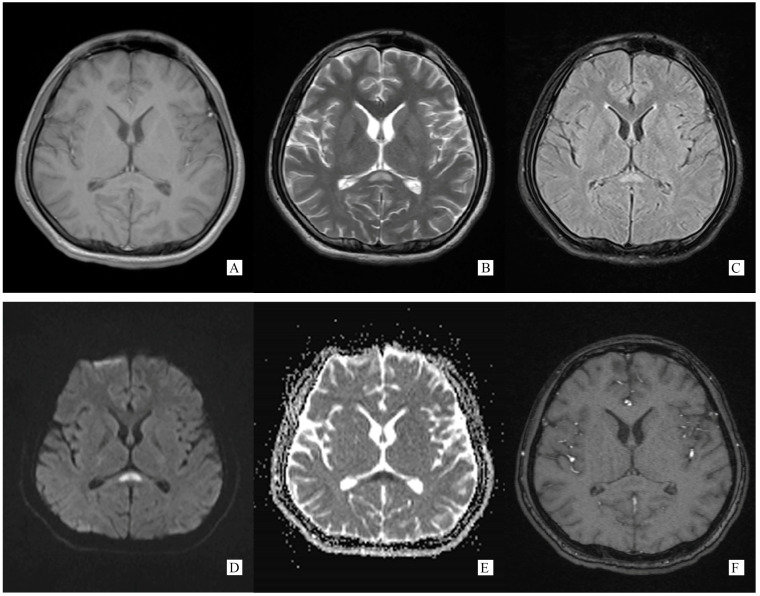
(Performed on 5 November 2021). (**A**) Axial T1 weight brain MRI image showing an oval hypointense lesion on SCC. (**B**) Axial T2 weight brain MRI image showing an oval hyperintense lesion on SCC. (**C**) Axial T2 flair weight brain MRI image showing an oval hyperintense lesion on SCC. (**D**) Axial DWI weight (b = 1000) brain MRI image showing an oval hyperintense lesion on SCC. (**E**) Axial ADC weight brain MRI image showing an oval hyperintense lesion on SCC. (**F**) Axial post-gadolinium T1-weighted brain MRI showing no enhancement on SCC lesion.

**Figure 2 brainsci-13-00659-f002:**
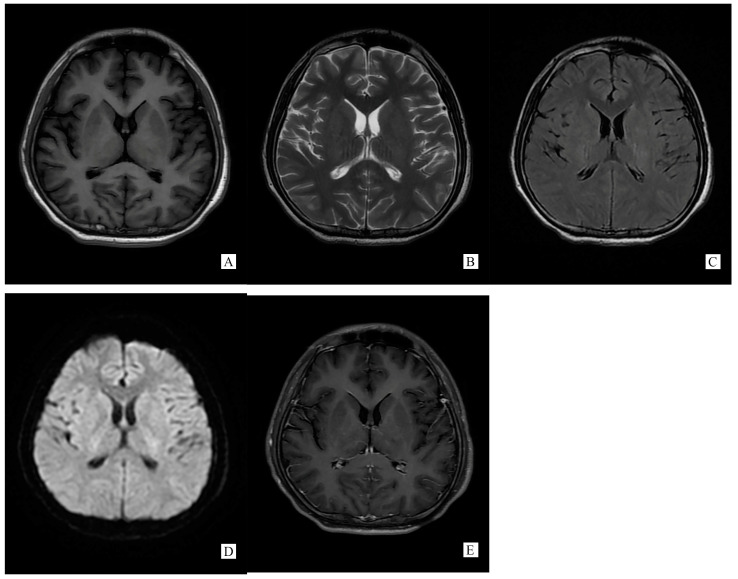
(Performed on 13 November 2021) SCC lesion totally disappeared on brain MRI ((**A**). T1WI, (**B**). T2WI, (**C**). T2 flair, (**D**). DWI, (**E**). Post-gadolinium T1-weighted).

**Table 1 brainsci-13-00659-t001:** Characteristics of the patient’s CSF.

CSF CharacteristicsTime	4 November 2021	8 November 2021	11 November 2021	17 November 2021	8 December 2021	11 May 2022
	Day7	Day11	Day14	Day17	Day41	Day195
ICP	350	80 drops/min	330 mmH_2_O	215 mmH_2_O	300 mmH_2_O	230 mmH_2_O
Appearance	clear and colorless	clear and colorless	clear and colorless	clear and colorless	clear and colorless	clear and colorless
WBC counts	0	206	500	240	12	16
Glu(mmol/L)	2.25	1.93	2.14	3.61	3.37	3.3
Cl^−^(mmol/L)	118.2	121.2	119.5	128	126	127.5
Pro(mg/L)	1270	1277.5	1051.9	499.2	307.6	337.7
ADA(U/L)	Unknown	8.5	8.8	6	2.7	2.2
CK(U/L)	Unknown	Unknown	12	8	Unknown	Unknown
LDH(U/L)	Unknown	Unknown	89	20	Unknown	Unknown
AST(U/L)	Unknown	Unknown	24	20	Unknown	Unknown
Ca^2+^(mmol/L)	Unknown	Unknown	1.1	1.1	Unknown	Unknown
Mg^2+^(mmol/L)	Unknown	Unknown	0.86	0.98	Unknown	Unknown
UA(μmol/L)	Unknown	Unknown	52	24	Unknown	Unknown
Alb(mg/L)	Unknown	Unknown	604.5	299	Unknown	Unknown
IgG(mg/L)	Unknown	Unknown	111.1	63.4	Unknown	Unknown
GFAP IgG	Unknown	Unknown	Positive (antibody titer 1:32)	Unknown	Unknown	Unknown
Oligoclonal bands	Unknown	Unknown	Positive (≥2 bands)	Unknown	Unknown	Unknown

**Table 2 brainsci-13-00659-t002:** Review of published cases of autoimmune GFAP autocytopathy coexist with RESLES.

Author, Year	Diagnosis	Flu-Like Symptoms	Pathogens	Neurological Symptoms	CSF Features	MRI Features	Treatment	Outcome
Oger et al., 2019 [5]	Autoimmune GFAP autocytopathy with RESLES	Exist	Not found	Progressively, alteration of consciousness, dysmetria, nystagmus, gait difficulties	WBC cell counts: 380 cells/mm^3^, proteins: 1.33 g/L, oligoclonal bands negative, GFAP antibody positive	Reversible SCC lesion with hyperintensity on T2 and DWI with increased ADC, disappeared within a week	IGIV infusion (1 g/kg/J) for 2 days	Completely recovered within 12 months
He’raud et al., 2021 [4]	Autoimmune GFAP autocytopathy with RESLES	Exist	Not found	progressive and worsening fatigue, headaches, diplopia, walking difficulties, multidirectional nystagmus, right side apraxia and anosognosia, multiple cranial nerve palsy (left sixth cranial nerve, right facial and bilateral trigeminal hypoesthesia), proprioceptive ataxia of the lower limbs	WBC cell counts: 121 cells/μL, proteins: 1.51 g/L, glucose: 1.93 mmol/L, GFAP antibody positive	Reversible SCC lesion with hyperintensity on T2 and diffusion sequences with hypointensity on T1, linear gadolinium enhancement of the Virchow-Robin perivascular spaces, SCC lesion disappeared within 29 days; hyperintensities on C2, C3, and C6 levels, diffuse leptomeningeal gadolinium enhancement	Started with amoxicillin and cepha losporin (worsening presented by altered consciousness, left facial paralysis, acute urinary retention, dysarthria, and complete visual loss in the left eye), then transferred to steroids therapy after diagnosed autoimmune GFAP autocytopathy	Completely recovered within 4 months
Fang et al., 2022 [6]	Primary central nervous system lymphoma coexistent with autoimmune GFAP astrocytopathy	No	Not found	Somnolence, memory declination	WBC cell counts: 12 cells/mm^3^, proteins: 0.455 g/L, GFAP antibody positive	Multiple abnormal enhancement lesions in bilateral basal ganglia and around the third ventricle, ands transient T2-weighted hyperintensity lesions at the SCC, SCC lesion attenuated in 2 months	Steroids therapy for about 2 months (worsening presented by poor response and even worsened clinical manifestations when the dose of prednisone reduced to 45 mg), then transferred to rituximab and methotrexate after diagnosed large B-cell lymphoma	Significantly improved after three courses of chemotherapy

## Data Availability

The data that support the findings of this study are available from the corresponding author upon reasonable request.

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
