# Peer review of "Autoimmune Glial Fibrillary Acidic Protein (Gfap) Astrocytopa-Thy Accompanied with Reversible Splenial Lesion Syndrome (RESLES): A Case Report and Literature Review"

_brainsci, 2023, doi:10.3390/brainsci13040659_

Round 1

Reviewer 1 Report

The manuscript Autoimmune glial fibrillary acidic protein (GFAP) astrocytopathy accompany with Reversible splenial lesion syndrome (RESLES): A case report and literature review. is well written and interesting. 

Only minor comments that could help to improve the paper. 

Title:

there is a discrepancy between authors title and affiliation

"Jingli Liu and PhD1 1 Department of Neurology, The First Affiliated Hospital of Guangxi Medical University, Nanning, China * Correspondence: Jingli Liu, M.D"

Results:

The legend of figure 1 and 2 are not homogenous. for figure 1 you note with lower cap "a", "b" and on the other you note with upper cap letter "A" "B". 

Also it would be useful to add the letter on each photo, and take care of the presentation, some photos are surrounded by a white frame, some are zoom in.  please improve the presentation of those 2 figures.

Discussion:

I suggest to discuss at a possible mechanism linking GFAP and RESLES. As well as the potential future direction science and clinicians should follow to improve the medical care and future treatments.

It would be interesting to provide a graphical abstract that summarizes both diseases.  

Reviewer 2 Report

The article entitled “Autoimmune glial fibrillary acidic protein (GFAP) astrocytopathy accompany with Reversible splenial lesion syndrome (RESLES): A case report and literature review” aims to describe a case report of a relationship between autoimmune GFAP astrocytopathy and RESLES which remains unclear and this can contribute for elucidation of it. This report is interesting and deserve the knowledge of scientific community. It will be helpful to create a table with resumed symptoms. Additionally, the author may give a perspective about treatment and complementary analysis that can be done. 

Author Response

Thank you for your advice. Actually we have already discussed the  treatment on the fifth paragraph of “Discussion”.

Reviewer 3 Report

Case report from Wang et al.  describing a potential case of autoimmune GFAP meningoencephalomyelitis that resolves with antiviral and steroid treatment.  This adds to the fairly short list of documented cases and is thus of interest to the community.

As a minor point, the authors may consider presenting some of the lab workup as a table in order for others to easily collate data as more cases become available.  

Author Response

Thank you for your advice. We have added “Table 1.” to summarize the lab workup.